# Evaluation of Interfacial Fracture Toughness and Interfacial Shear Strength of Typha Spp. Fiber/Polymer Composite by Double Shear Test Method

**DOI:** 10.3390/ma12142225

**Published:** 2019-07-10

**Authors:** Samsul Rizal, Yoshikazu Nakai, Daiki Shiozawa, H.P.S. Abdul Khalil, Syifaul Huzni, Sulaiman Thalib

**Affiliations:** 1Department of Mechanical Engineering Syiah Kuala University, Darussalam, Banda Aceh 23111, Indonesia; 2Department of Mechanical Engineering Kobe University, 1-1, Rokkodai, Nada, Kobe 657-8501, Japan; 3School of Industrial Technology, University Sains Malaysia, Penang 11800, Malaysia

**Keywords:** *Typha spp.* fiber, surface treatment, interfacial fracture, interfacial shear

## Abstract

The aim of this paper is to evaluate the Mode II interfacial fracture toughness and interfacial shear strength of *Typha spp.* fiber/PLLA and *Typha spp.* fiber/epoxy composite by using a double shear stress method with 3 fibers model composite. The surface condition of the fiber and crack propagation at the interface between the fiber and the matrix are observed by scanning electron microscope (SEM). Alkali treatment on *Typha spp.* fiber can make the fiber surface coarser, thus increasing the value of interfacial fracture toughness and interfacial shear strength. *Typha spp.* fiber/epoxy has a higher interfacial fracture value than that of *Typha spp.* fiber/PLLA. Interfacial fracture toughness on *Typha spp.* fiber/PLLA and *Typha spp.* fiber/epoxy composite model specimens were influenced by the matrix length, fiber spacing, fiber diameter and bonding area. Furthermore, the interfacial fracture toughness and the interfacial fracture shear stress of the composite model increased with the increasing duration of the surface treatment.

## 1. Introduction

The use of natural fibers as reinforcement of polymers composite has been researched by many scientists and has grown rapidly [1,2,3]. At present, the availability of resources and environmental aspects has become one of the biggest challenges [4]. The green composite concept has attracted the interest of many researchers and engineers due to the superiority of these composites in terms of mechanical performance and excellency in energy efficient and environmental aspects such as bio-degradable, renewable and recyclable [5,6,7,8,9]. The all green composites promote the utilization of composites consisting polymers and reinforcement materials from renewable resources. Poly (L-lactide acid) (PLLA) is a polymer derived from renewable resources, and often used as a matrix for composite manufacturing [10,11]. Meanwhile, *Typha spp.* fiber is one of the natural fibers that can be considered as a polymer reinforcement for composite [12]. *Typha spp.* is a plant resembling reed that wildly grows in waters and swamps throughout the world, and its adaptability to climate changes and various environmental conditions causes the rampant growth of *Typha spp*. The widespread of Typha dominates the area and raises concerns for the community as it is cannot be cultivated nor exploited well. This makes *Typha spp*. being known as a parasitic plant [12,13]. *Typha spp.* plant is presented in Figure 1.

Despite its merits for environmental friendliness, many studies reported that the addition of natural-sourced fiber as a reinforcement can reduce the mechanical performance of the natural-fiber-reinforced composites due to the weak interface adhesion ability between the fiber and the matrix [14,15,16,17]. The mechanical properties of composites depend on the strength of the fiber, and the matrix and the adhesion between the fiber and the matrix. The interface adhesion is the conductor of stress from the matrix to the fiber [18,19]. The weak interface adhesive between the natural fiber and the polymer matrix is attributed to the difference in wettability properties of the natural fiber which is hydrophilic and the polymer matrix which is hydrophobic. This phenomenon reduces the efficiency of the stress distribution from the matrix to the fiber and simultaneously diminishes the mechanical properties of the composite [20]. Generally, natural fiber is hydrophilic with the main components of natural fiber being cellulose, hemicellulose and lignin. Cellulose is the imperative structural component in natural fiber, and it is a semi-crystalline polysaccharide. Hemicellulose was a cementing matrix between cellulose and lignin which give rigidity to plants. While cellulose and hemicellulose are hydrophilic, lignin is relatively hydrophobic [21]. 

The interface bonding of the fiber and the matrix occurs through electrostatic and chemical bonding as well as mechanical interlocking mechanisms [22]. Mechanical interlocking occurs when the surface of the fiber is rough. This mechanism increases the shear strength of the fiber interface and the matrix. It is possible that several types of bonding between the fiber and the matrix interfaces occur simultaneously [23]. Due to the poor compatibility of natural fibers with polymers, treatment is needed to modify the structure of the fiber’s surface and eliminate hemicellulose and lignin from the fiber. Alkali treatment is one of the most common treatments used for natural fibers. Apart from reducing the lignin, wax, pectin, oil, and hemicellulose components on the surface of the fiber, alkali treatment can also interfere with the hydrogen bonds in the chemical structure of natural fibers and increase fiber surface roughness. Moreover, alkali treatment has increased the crystallinity index of the fiber, thus the formulation of hydrogen bonds of the cellulose chains enhanced the chemical bonding between the fibers in composites [24,25,26]. However, it should be noted that the treatment with high alkali concentrations and long treatment periods will reduce the mechanical properties of the fibers [27,28]. 

Since the interface bonding between the fiber and the matrix plays an important role in the strength of natural-fiber-reinforced composites, there is an increasing interest in developing test methods to study the behavior of composite fractures, especially in interfacial fracture toughness which affects the strength of composite materials at the macroscopic level. Several common test methods currently used to evaluate interfacial fractures between the fibers and the matrix are fiber pull-out tests [29], fiber push-out [30], the microbond test [31,32,33] and Broutman test [34]. The crack propagation in fiber-reinforced composites was evaluated from the fracture process that occurs at the interface of fiber and matrix [35,36,37]. The crack propagation behavior on fiber-reinforced composites was strongly influenced by the interface between the fiber and the matrix as reported by Kotaki et al. [38] and Hojo et al. [39]. However, the evaluation of interfacial fracture toughness by using a real size composite model can be measured by Mode I/Mode II crack growth as examined by Koiwa et al [40], who reported that the increasing bonding length caused an improved interfacial fracture toughness. However, the results of the measurement of fracture toughness were obtained in a very short matrix size range.

The aim of this work is to evaluate the Mode II interfacial fracture toughness and interfacial shear strength of *Typha spp.* fiber/PLLA and *Typha spp.* fiber/epoxy composite by using a double shear stress method with a three-fiber model composite. The surface of fiber conditions and crack propagation at the interface between the fiber and the matrix are observed by the scanning electron microscope (SEM). Two parameters are employed to evaluate the interfacial fracture toughness, one is energy release rate, *G*, and the other is the mean interfacial shear stress, τ. When the singular stress field due to cracking prevails over the fracture process zone, the condition of the fracture is determined by the critical value of *G*. Otherwise, the fracture process zone prevails over the sample, and the condition is determined by the critical value of τ.

## 2. Materials and Methods 

### 2.1. Materials 

*Typha spp.* fiber was collected from the swamp areas of the Syiah Kuala sub-district, Banda Aceh, Indonesia. The stem of *Typha spp.* was decorticated to obtain the fibers. The fibers were dried by sun-drying. Some fibers were soaked in 5% NaOH solution for 1, 2, 4, and 8 h. After alkali immersion, the fibers were washed with water to stop the alkali process. Following that, the fiber was dried at room temperature for 48 h and then used as a composite sample [41]. The fiber was prepared with a diameter of approximately 129–561 μm. The sodium hydroxide (NaOH) (Fujifilm Wako Pure Chemical Corp., Japan) was acquired in Hyogo, Japan. Poly (l-lactide acid) fibers were obtained from Unitika Ltd., Osaka, Japan with a 200 µm diameter, and epoxy resin was supplied by Epoch Corp., Osaka, Japan.

### 2.2. Scanning Electron Microscopy (SEM)

The surface of untreated and alkali-treated *Typha spp.* fibers, and the crack propagation of interfacial fracture toughness were inspected by using a scanning electron microscope (Hitachi TM3000, Tokyo, Japan). The diameter of the fiber, matrix length, spacing fiber and bonding angle, 2θ were observed by a Hirox KH-1300 digital microscope, Tokyo, Japan.

### 2.3. Mode II Interfacial Fracture Toughness Test 

The double shear test method was conducted to determine the value of the interfacial fracture toughness of *Typha spp.* fiber/PLLA and *Typha spp.* fiber/epoxy. The schematic of the three-fiber model composite specimens is shown in Figure 2. The three-fiber model composite was prepared for the Mode II interface test. Before preparing the specimen, *Typha spp.* fibers were affixed in parallel to the paper with a slight pull to keep it tight. Epoxy glue was used to adhere both ends of the fiber. The epoxy-composite sample was obtained by dropping epoxy resin on the three arranged fibers. The epoxy resin was mixed with hardener and left to stand until the texture of the epoxy became jelly-like to facilitate the attachment of epoxy resin to the fiber, because the epoxy resin was originally in liquid form. Once the epoxy resin cured, the matrix length of the specimen was measured. For the *Typha spp.* fiber/PLLA composite specimen, PLLA was melted in between the three *Typha spp.* fibers. The matrix length was measured after the melted PLLA has solidified completely. Hereafter, a mechanical test was carried out using Tohei MT201 tensile machine (Tokyo, Japan) with a capacity of 50N under cross-head speed of 0.15 mm/s. The composite model which mounted on the paper tab was carefully placed in the middle of the 2 clamps. Both sides of the marker line were cut carefully after the clamp was tightened to prevent fracture before the test started.

The strain energy of the model composite specimen can be calculated by adapting the mathematical model used by Sia et al [42], as follow:(1)$$U=U_{i\;}+U_{\mathrm{ii}}+U_{\mathrm{iii}}$$

The *U*_i_, *U*_ii_ and *U*_iii_ are the strain energy of the section (i), (ii), and (iii) in Figure 1. The equations are stated below.
(2)$$\begin{array}{l} U_{i}=\frac{2P^{2}n}{\pi ED^{2}}\\ U_{i}{}_{i}=\frac{2P^{2}L}{3\pi ED^{2}}\\ U_{i}{}_{\mathrm{ii}}=\frac{P^{2}m}{\pi ED^{2}} \end{array}$$
where the Young’s Modulus, *E* = 0.88 GPa for untreated fiber, 1.55 GPa for alkali-treated 1 h, 1.62 GPa for alkali-treated 2 h, 1.47 GPa for alkali-treated 4 h and 1.23 GPa for alkali-treated 8 h. The fiber Young’s modulus was determined by a single bundle fiber tensile test according to ASTM D3379-75 standard. *D* = Diameter of *Typha spp.* fiber, *P* = applied force, *L* = matrix length, *m* = left fiber length and *n* = right fiber length.

The energy release rate at crack tip A and B are given by the equations below.
(3)$$G_{A}=\frac{1}{2\theta D}\cdot \frac{d}{dm}(U_{i}+U_{\mathrm{ii}}+U_{\mathrm{iii}})=\frac{1}{2\theta D}\left(\frac{dU_{i}}{dm}-\frac{dU_{\mathrm{ii}}}{dL}\right)=\frac{P^{2}}{6\pi ED^{2}\theta }$$
(4)$$G_{B}=\frac{1}{2\theta D}\cdot \frac{d}{dn}(U_{i}+U_{\mathrm{ii}}+U_{\mathrm{iii}})=\frac{1}{2\theta D}\left(\frac{dU_{i}}{dn}-\frac{dU_{\mathrm{ii}}}{dL}\right)=\frac{2P^{2}}{3\pi ED^{2}\theta }$$

The interfacial shear strength (IFSS) of the composite specimen can be obtained with the bonding area, $A=D\theta L_{\min}$. Where *L*_min_ is the outer fiber bonded with the minimum matrix length. The mean interfacial shear stress can be obtained as:(5)$$\tau_{i}=\frac{P}{2D\theta L_{\min}}$$

In the fabrication process, the equal length of the matrix on both sides of the specimen is very difficult to achieve. Therefore, the average length of both sides of the matrix was used as the matrix length, *L*. The length of the fiber, *m*, and *n* were also obtained from the mean length of both sides of the specimen. The average fibers spacing, *t* was obtained from 10 random measurements of the distance between the fiber specimens. The schematics of the model composite specimen after fabrication are displayed in Figure 3, and the condition of composite specimen on the test equipment is shown in Figure 4.

## 3. Results and Discussion

### 3.1. Surface Morphology of Typha Spp. Fiber

The evaluation of the surface morphological of the untreated and the alkali-treated *Typha spp.* fibers was performed a by scanning electron microscope. The results are displayed in Figure 5. From the SEM imaging result in Figure 5a, it is clearly visible that the external surface of the fiber cell wall of the untreated *Typha spp.* fiber is still covered by impurities such as hemicellulose, lignin and wax. Meanwhile, for the 1 h alkali-treated sample (Figure 5b), the surface impurities are seen to be slightly reduced. Increasing the immersion time in alkali to 2, 4, and 8 h further reduced the presence of impurities on the Typha fiber, as shown in Figure 5c–e. Furthermore, alkali treatment also reduced the diameter of the fiber. This occurrence was similar to that of reported by Obi Reddy et al. on alkali-treated Borassus fruit fine fibers with 5% alkali treatment solution [43]. The average Typha fiber diameters after alkali treatment are presented in Table 1. 

### 3.2. Interfacial Fracture Toughness of the Typha Spp. Fiber Model Composite

The interfacial fracture toughness, *G*_i_ of the untreated and 5% alkali-treated *Typha spp.* fiber/PLLA model composites are presented in Figure 6 as a function of (a) matrix length, *L*, (b) fiber spacing, *t*, (c) fiber diameter, *D*, and bonding area, *A*. Since the results are not expressed by a single parameter, the least square regression was conducted to fit the following equation:(6)$$G_{i}=G_{i0}+k_{L}L+k_{t}t+k_{D}D+k_{A}A\;$$

To check the validity of the least square regression, the measured value of *G*_i_ of each sample was plotted as a function of the least square regression value of *G*_i_ which was calculated from Equation (6) by substituting measured values of *L*, *t*, *D*, and *A* of each sample in Figure 7. Although a certain amount of scatter exists, no systematic deviation was found. Then, the linear combination of each factor indicated by Equation (6) is considered to be valid in the range of the present experiments, and the real scatter of the interfacial fracture toughness is shown the figure.

The lines in Figure 6 show the effect of each parameter given by Equation (6) where other parameters are fixed to the average values. As seen from Figure 6a–d, *G*_i_ values increased with the increasing matrix length and fiber spacing, and *G*_i_ values were found to decrease with increasing fiber diameter and bonding area. In all cases, the untreated *Typha spp.* fiber model composite had less interfacial fracture toughness value than the alkali-treated fibers. This phenomenon occurs because the alkali treatment changes the surface roughness contour of the *Typha spp.* fiber. The surface of the alkali-treated fibers is rougher than the raw fiber due to the removal of surface impurities by the alkali treatment [44]. Since the fiber becomes cleaner from the impurities and the surface becomes coarser, the fiber-matrix mechanical interlocking bonding occurs and promotes better adhesion between the fiber and the matrix.

In the case of interfacial fracture toughness, from the *G*_i_ versus the matrix length plot shown in Figure 6a, greater values of interfacial fracture toughness were observed at longer matrix lengths. The greater the length of the matrix, the higher the interfacial fracture toughness becomes. This finding is similar to the results reported by Chin Von Sia [42]. The highest *G*_i_ value obtained for the untreated fiber was only 27.4 J/m^2^, which was much smaller than the 1 h, 2 h, 4 h, and 8 h alkali-treated *Typha spp.* fibers whose *G*_i_ value were 78.1, 96.1, 91.4, and 86.2, respectively. The phenomenon of the *G*_i_ value increment was also found in the *G*_i_ versus fiber spacing plot presented in Figure 6b. The gap of fibers on the *Typha spp.* fiber/PLLA model composite specimen affects the *G*_i_ value. The greater gap between the fibers increased the G_i_ value of *Typha spp.* fiber/PLLA model composite. The plotting results of *G*_i_ versus the fiber diameter in Figure 6c, and *G*_i_ versus the bonding area in Figure 6d showed the distribution of the *G*_i_ value decreased. 

Figure 8 shows the trends of interfacial fracture toughness, *G*_i_ values of *Typha spp.*/epoxy model composite on/against matrix length (Figure 8a), fiber spacing (Figure 8b), fiber diameter (Figure 8c), and the bonding area (Figure 8d). The lines in each figure show the relationship obtained using the least square regression employing the same manner as Figure 7. Similar to Figure 7, no systematic deviation was found in the relationship between the measured and least regression values of *G*_i_ as shown in Figure 9. This indicates the validity of the linear combination of each factor in Equation (6).

Similar to the results in Figure 6, the distribution of the *G*_i_ values were found to be increased against the matrix length and the fiber spacing, and decreased on the fiber diameter and the bonding area. However, it can still be seen that, in general, the *G*_i_ of untreated *Typha spp.* fiber is lower when compared to the alkali-treated *Typha spp.* fibers. This indicates that the value of the interfacial fracture toughness of the model composite *Typha spp*./PLLA and *Typha spp*./epoxy exhibit the same behavior. 

The comparison of the mean interfacial fracture toughness between *Typha spp.* fiber/PLLA and *Typha spp.* fiber/epoxy is shown in Figure 10. Figure 10 shows that the *Typha spp.* fiber/epoxy model composite has higher interfacial fracture values than *Typha spp.* fiber/PLLA model composite. This occurs because the PLLA-based composite has lower mechanical properties compared to the epoxy. The PLLA has poor melt strength, brittleness and the melt viscosity of PLA has low shear sensitivity and relatively poor strength [45,46]. The value of interfacial fracture toughness on alkali-treated *Typha spp.* fiber increased. This happens because the surface of the fiber becomes rougher. Sangappa et al reported that the surface roughness of natural fibers increases with increasing duration of alkaline treatments [41]. The highest value was found on alkali-treated *Typha spp.* fiber/epoxy for 8 h with 83.9 J/m^2^, and the lowest value was on the *Typha spp.* fiber/PLLA with 14.4 J/m^2^.

### 3.3. Interfacial Shear Strength of the Typha Spp. Fiber Model Composite

The interfacial shear stress, *τ*_i_ of *Typha spp.* fiber/PLLA model composite and its relationship with specimen parameters is plotted in Figure 11, where lines indicate the relationship obtained by the least square regression employing average values of other factors of the abscissa. In this case, no systematic deviation was found in the relationship between the measured and the least regression values of τ_i_ as shown in Figure 12. Figure 11a illustrates the correlation of interfacial shear stress, *τ*_i_ with matrix length, *L*. The results showed that the interfacial shear stress, *τ*_i_ of the *Typha spp.* fiber/PLLA model composite decreased with increasing matrix length. A similar trend was also seen in Figure 11d in which the interfacial shear strength value decreased with the increasing bonding area. Similar results were reported by Day and Cauich Rodrigez in a study on Kevlar fiber [31], Zhandarov and Mader on glass fiber [47], and Gorbatkina et al. on carbon fiber [48] with thermosetting or thermoplastic as the matrix. The increasing bonding area enhanced the occasion probability of the critical flaws, therefore the probability of the initiated fracture upon loading was increasing [49]. Otherwise, the interfacial shear stress values were increased against the average fiber spacing and fiber diameter as shown in Figure 11b,c.

The same phenomenon occurs in the *Typha spp.* fiber/epoxy model composite specimens where the value of interfacial shear stress decreases with the increasing matrix length and bonding area where lines indicate the relationship obtained by the least square regression employing average values of other factors of the abscissa. As illustrated in Figure 13a,d. In this case, also no systematic deviation was found in the relationship between the measured and the least regression values of τ_i_ as shown in Figure 14. On the other hand, the data for interfacial shear stress versus fiber spacing (Figure 13b) and fiber diameter (Figure 13c) increased. In addition, the higher value of the interfacial shear stress was observed on the alkali-treated *Typha spp.* fiber/epoxy specimens due to the increased surface roughness and the lesser amount of hemicellulose, lignin and wax content on the fiber by alkali treatment. This promotes a better fiber-matrix bonding condition and improves their interface adhesion. Boopathi et al. also reported that strong hydrogen bonds seen in alkali-treated fibers facilitated better mechanical properties for interfacial adhesion between the fiber and the matrix [44].

The average interfacial shear strength of *Typha spp.* fiber/PLLA and *Typha spp.* fiber/epoxy is shown in Figure 15. It is seen that the average interfacial shear strength in the alkali-treated *Typha spp.* fiber is higher than in the untreated one. This is attributed to the low compatibility of the hydrophobic polymer-based matrix with the hydrophilic untreated fiber due to the presence of pectin and wax substances which act as a barrier for natural fibers to interlock with the polymer matrix [21].

To study the effect of difference matrix component, the interfacial shear strength results of both *Typha spp*.-based composites were compared and shown in Figure 15. The highest value was found on both model composites which *Typha spp.* fiber was alkali-treated for 2 h. However, a decreasing trend of the interfacial shear strength was seen on the *Typha spp.* fibers which underwent alkali treatment for 4 h and 8 h. This decrease occurred because the structures of the fibers were damaged after long-term alkali immersion [14,43]. The interfacial shear strength of 4 h and 8 h alkali-treated *Typha spp.* fiber/PLLA composites specimens were slightly higher than the *Typha spp.* fiber/epoxy. This is because *Typha spp.* fiber/PLLA has a physical interaction mechanism between the fibers and the polymers. On the contrary, the interface adhesion of *Typha spp.* fiber/epoxy is mainly formed through chemical bonding [50].

The micrograph of the fracture surface at the initial crack tip of the alkali-treated *Typha spp.* fiber/PLLA and *Typha spp.* fiber/epoxy composite models is shown in Figure 16. The crack is propagated on the outer surface of fiber and matrix where the surface of the fiber can be clearly observed. The damage occurs on the surface of the fiber which causes the surface of the fiber to peel off.

### 3.4. Effects of Geometrical Factor on Interfacial Fracture Toughness

In the previous discussion, the fracture condition obtained by the double shear test was not a unique critical value of the energy release rate, *G*, or the mean interfacial shear stress, τ, because the size of the singular stress is comparable to the size of the microstructure, such as the matrix length, *L*, fiber spacing, *t*, fiber diameter, *D*, or bonded area, *A*. However, in every case, the critical value of *G*, increased with the increasing *L* and *t*, and decreased with the increasing *D* and *A*, while the critical value of τ increased with *t* and *D*, and decreased with *L* and *A*. Then, *t* and *A* brought the same, and *L* and *D* had a counter effect for *G* and τ. The volume of resin between fibers increased with the increasing *L*, *t*, and *A*, and decreasing with *D.* This volume is a concerning amount of plastic deformation. On the other hand, the interface area at the notch tip increased with *A*. The effects of factors, *L*, *t*, *D*, and *A*, the effect of energy dissipation by plastic deformation and the statistical character of the interface strength should be considered. For the quantitative analysis of the effect of these factors, numerical analysis using finite element method can be conducted.

## 4. Conclusions

The Mode II interfacial fracture toughness and interfacial shear strength of *Typha spp.* fiber/PLLA and *Typha spp.* fiber/epoxy composite by using a double shear stress method was evaluated. The results obtained are summarized as follows

Alkali treatment on *Typha spp.* fiber can make the fiber surface coarser due to the removal of impurities, such as fatty substance from the fiber surface, thus increasing the interfacial fracture toughness and interfacial shear strength values.The *Typha spp.* fiber/epoxy has a higher interfacial fracture value than *Typha spp.* fiber/PLLA because the PLLA-based composite has lower mechanical properties compared to the epoxy, the PLLA has poor melt strength, brittleness and the melt viscosity of PLA has low shear sensitivity and relatively poor strength.Interfacial fracture toughness of *Typha spp.* fiber/PLLA and *Typha spp.* fiber/epoxy composite model specimens are influenced by the matrix length and the fiber spacing. The longer the matrix length, the higher the value of the interfacial fracture toughness. Meanwhile, the interfacial shear strength decreases with the increasing matrix length and the bonding area. Furthermore, the interfacial fracture toughness and the interfacial shear stress of the composite model increased with the increasing duration of the surface treatment.

## Figures and Tables

**Figure 1 materials-12-02225-f001:**
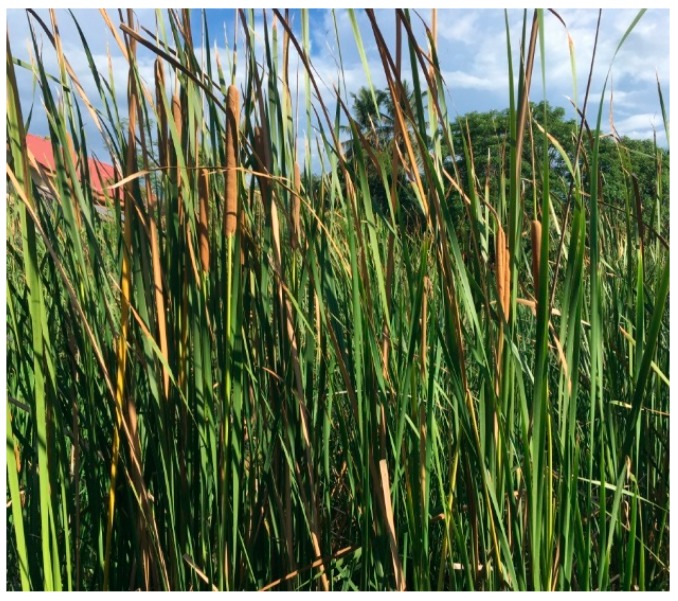
Typha spp. plant.

**Figure 2 materials-12-02225-f002:**
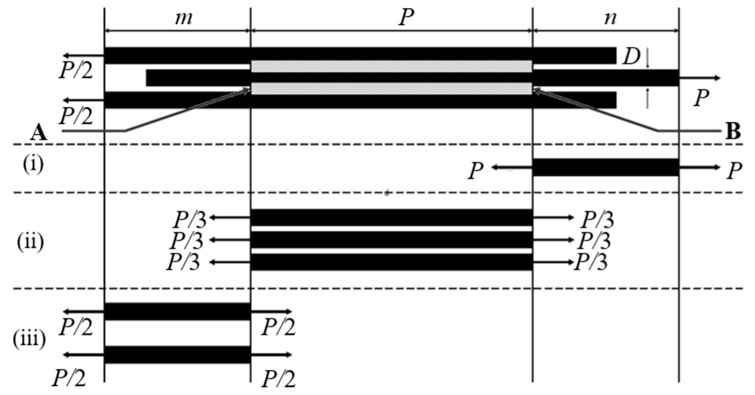
Schematic of the double shear method.

**Figure 3 materials-12-02225-f003:**
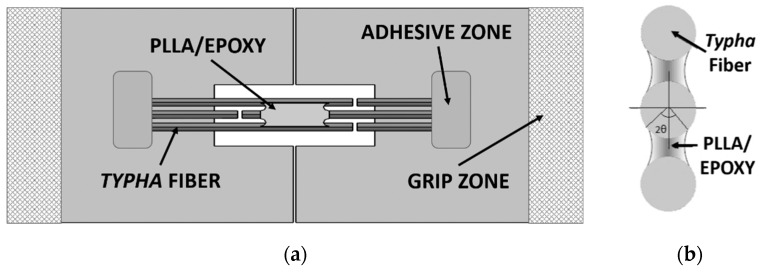
Schematics of the model composite. (**a**) Top view, (**b**) cross-section.

**Figure 4 materials-12-02225-f004:**
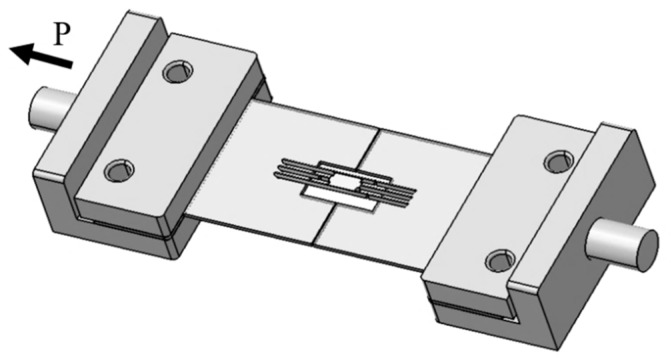
Schematics of the double shear method of the three-fiber model composite.

**Figure 5 materials-12-02225-f005:**
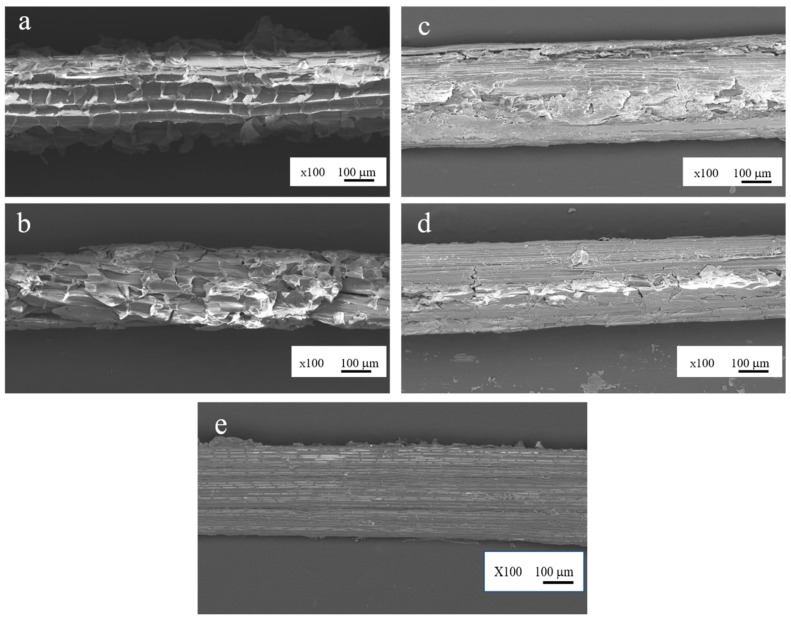
Surface imaging of (**a**) untreated, (**b**) Alkali-treated for 1 h, (**c**) Alkali-treated for 2 h, (**d**) Alkali-treated for 4 h, and (**e**) Alkali-treated for 8 h *Typha spp*. fibers.

**Figure 6 materials-12-02225-f006:**
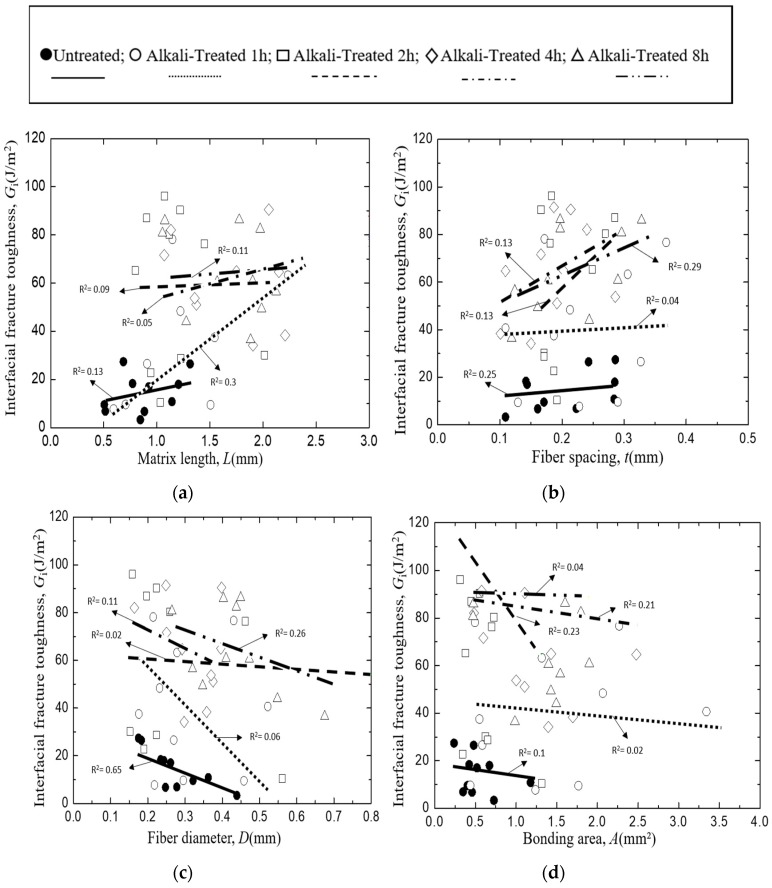
Interfacial fracture toughness of *Typha spp.*/PLLA, where: (**a**) Matrix length, (**b**) Fiber spacing, (**c**) Fiber Diameter, (**d**) Bonding area.

**Figure 7 materials-12-02225-f007:**
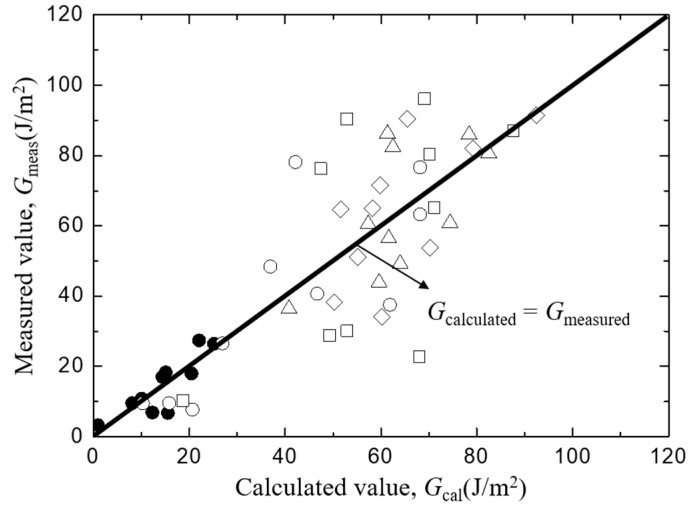
The comparison between measured and calculated interfacial facture toughness of *Typha spp.*/PLLA.

**Figure 8 materials-12-02225-f008:**
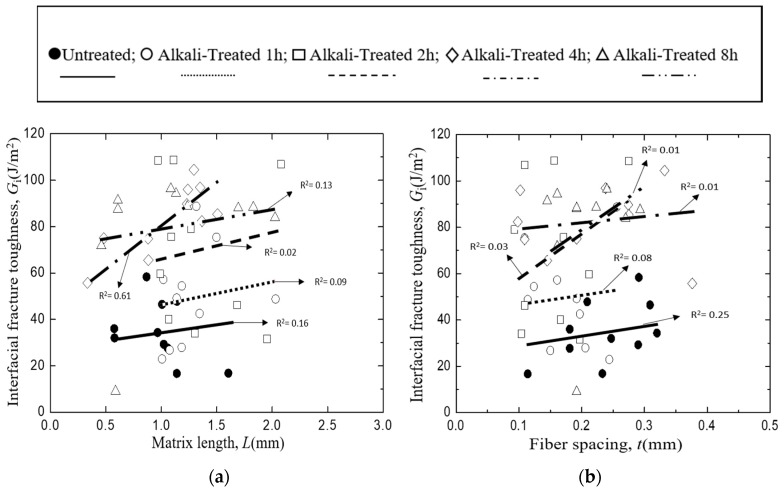
Interfacial fracture toughness of *Typha spp.*/Epoxy, where: (**a**) Matrix length, (**b**) Fiber spacing, (**c**) Fiber Diameter, (**d**) Bonding area.

**Figure 9 materials-12-02225-f009:**
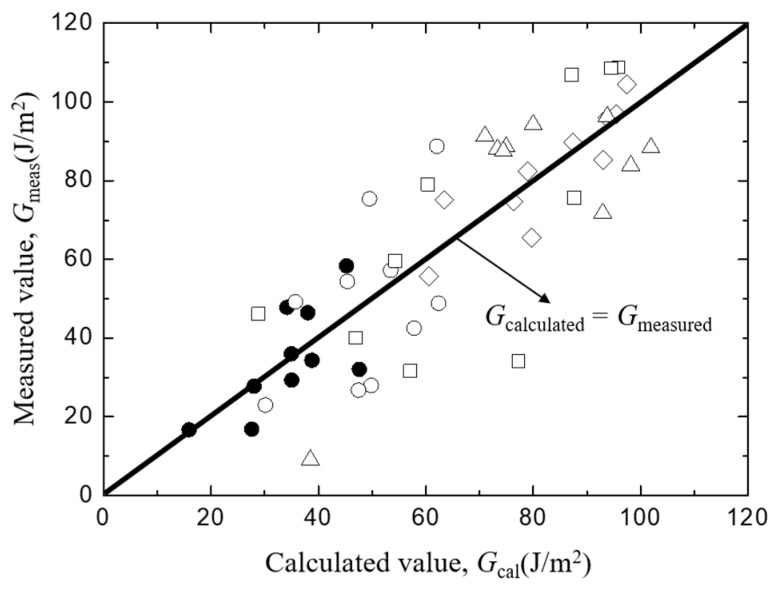
The comparison between measured and calculated interfacial fracture toughness of *Typha spp.*/Epoxy.

**Figure 10 materials-12-02225-f010:**
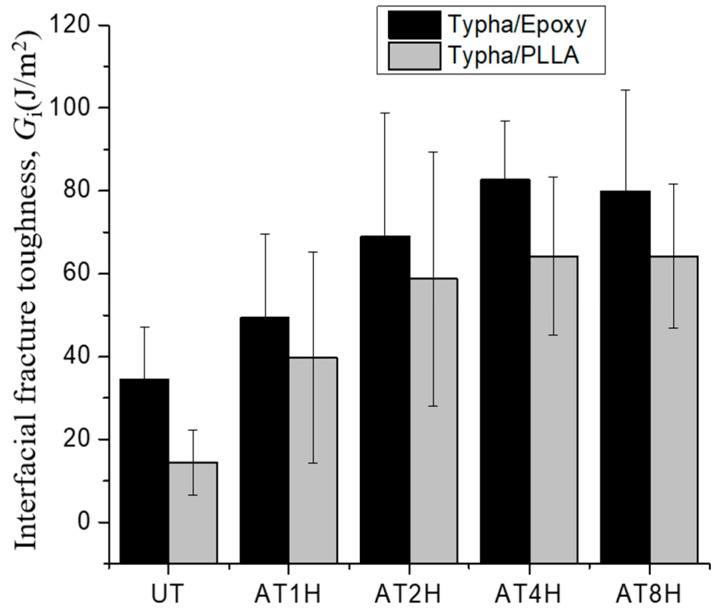
The comparison of the mean interfacial fracture toughness *Typha spp.* fiber/PLLA and *Typha spp.* fiber/Epoxy.

**Figure 11 materials-12-02225-f011:**
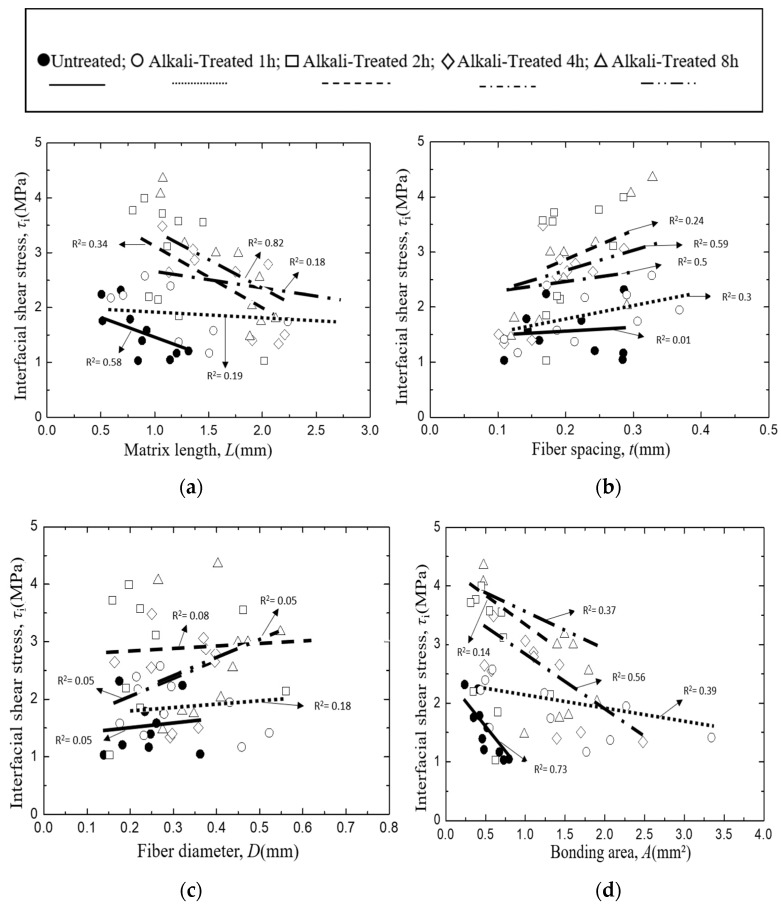
Interfacial shear strength of *Typha spp.*/PLLA, where: (**a**) Matrix length, (**b**) Fiber spacing, (**c**) Fiber Diameter, (**d**) Bonding area.

**Figure 12 materials-12-02225-f012:**
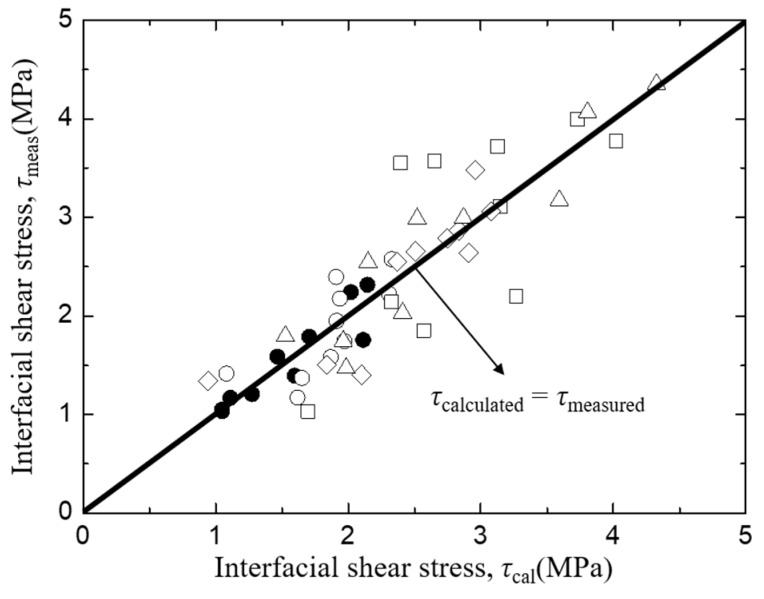
The comparison between measured and calculated interfacial shear strength of *Typha spp.*/PLLA.

**Figure 13 materials-12-02225-f013:**
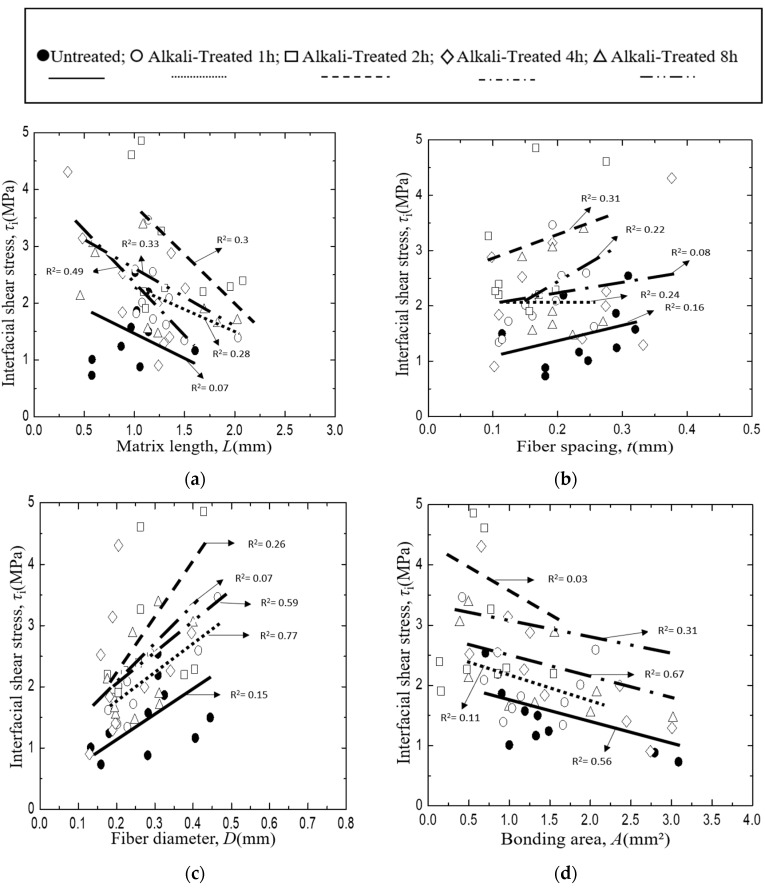
Interfacial shear strength of *Typha spp.*/Epoxy, where: (**a**) Matrix length, (**b**) Fiber spacing, (**c**) Fiber Diameter, (**d**) Bonding area.

**Figure 14 materials-12-02225-f014:**
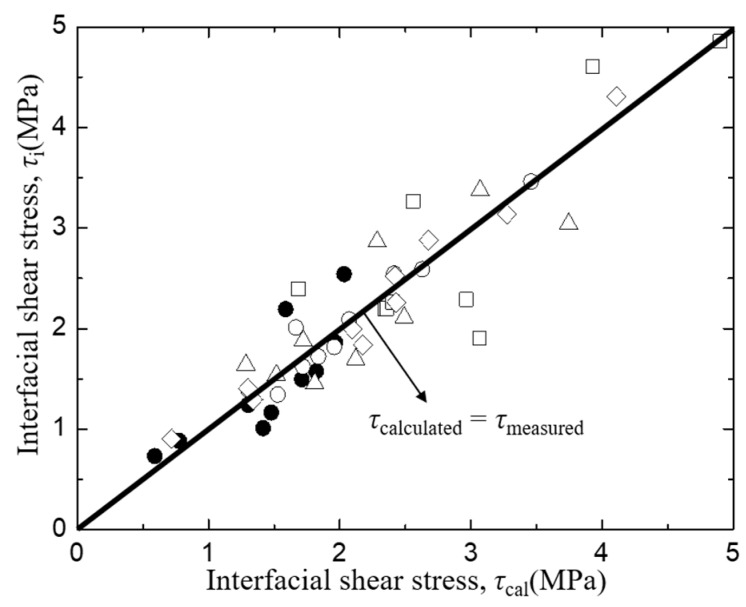
The comparison between the measured and calculated interfacial shear strength of *Typha spp.*/ Epoxy.

**Figure 15 materials-12-02225-f015:**
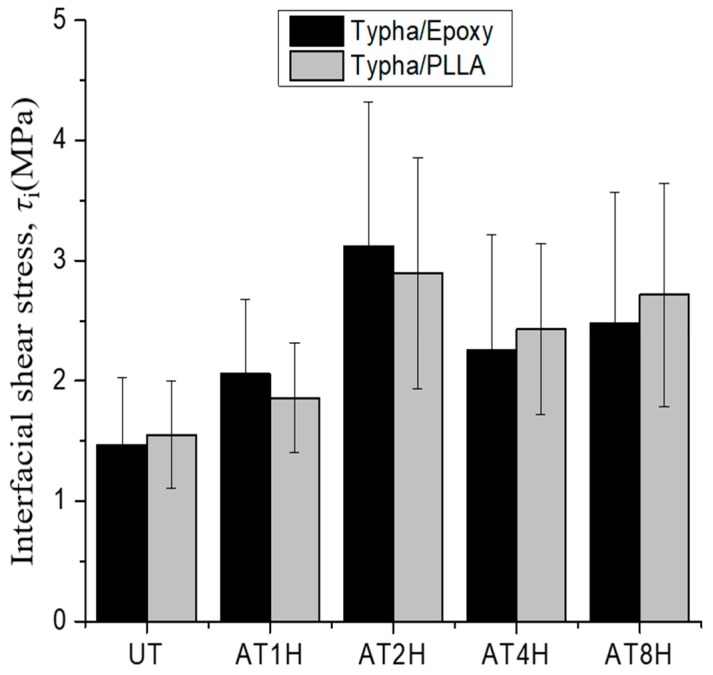
The comparison of the mean interfacial shear strength of *Typha spp.*/PLLA and *Typha spp.*/Epoxy.

**Figure 16 materials-12-02225-f016:**
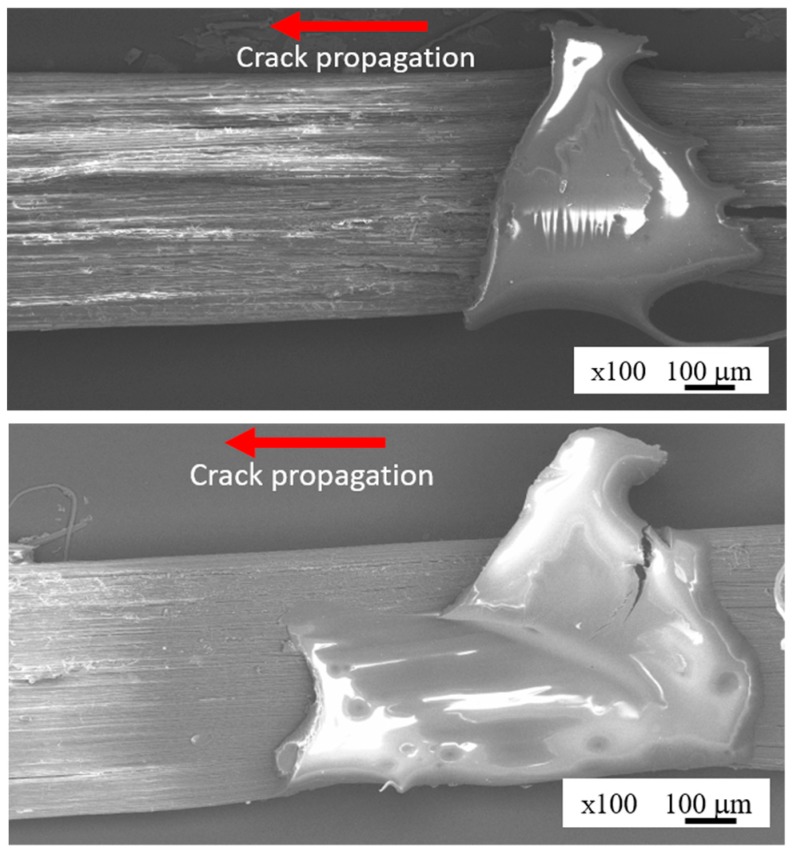
The specimen fracture surface.

**Table 1 materials-12-02225-t001:** The diameter of untreated and alkali treated *Typha spp*. fiber.

Alkali Treatment Period	Average Diameter (μm)
0	309
1	286
2	282
4	263
8	226

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
