# Peer review of "Evaluation of Interfacial Fracture Toughness and Interfacial Shear Strength of Typha Spp. Fiber/Polymer Composite by Double Shear Test Method"

_materials, 2019, doi:10.3390/ma12142225_

Round 1
Reviewer 1 Report
The manuscript presents a well-prepared basic research. The applied test methods and procedures meet the desired goals. However, the manuscript needs many modifications, after which it can be published.
The abstract from the line 22 to the line 28 is same like the conclusion from the line 330 to the line 338. This is not correct. In the conclusion must be explain the cause effect. Please change the conclusion!
In the line 70 there is a missing space: „… of the fibers[24,25].” Please correct it!
In the line 102 the authors gave the diameter of fibers in mm (0.2 mm) but it would be better if give it in µm (200 µm). Please change it!
In the line 119 the authors wrote: „… the melted PLLA has cured completely.” It would be more correct if use solidified instead of cured in case of thermoplastics.
In the line 133 the authors used only one equation number (2) but there are three different equation in this line. Please modify them and use three different equation number each!
In the line 134 the authors gave the values of E in every types of fiber. What is E? Young modulus? Please name it!
In the line 148 the authors wrote „The average distance between fibers, t was obtained …” but later the authors use t as fiber spacing. It would be better if introduce this name here or use only one expression for t.
In Fig. 5. the authors wrote „… and (e) Alkali-treated for 8h …” but the (e) picture is missing in the figure. Please replace it!
In the line 197 the authors used J/mm2 as unit of interfacial fracture toughness, but everywhere else they used J/m2. Which is the real unit? Please use unified units!
It would be better if use same axis subtitles in case of X axis on Fig. 6(a) and Fig. 11(a) and on Fig. 6(d) and Fig. 11(d). Please modify it!
In Fig. 7., Fig. 9. and Fig. 12. the authors used Gevaluated notations instead of Gcal which means calculated value. Please change it in every figures!
In Fig. 10. and Fig. 15. the label of Y axes are missing. Please replace them!
In the first sentence in section 3.4 (in the lines 314 and 315), the introduction of G is in wrong place, because G means fracture toughness and not energy release rate. Please change it!
In the line 315 there is a misspelling “… the energy release rare …” but the correct word is rate. Please change it!
The sentence from the line 317 to the line 319 (“However, in every case …”) is not correct. It would be good to explain the direction of geometric changes, instead of : “… the critical value of G, increase with L and t, and decrease with D and A …”, use: the critical value of G increase with increasing with L and t and decrease with increasing with D and A.
Reviewer 2 Report
1) Page 2 Line 56-57 “Natural fibers are hydrophilic due to the presence of hemicellulose and lignin which weakens the bonds between fibers and matrix”.
The statement is not fully right. In natural fibers, cellulose, the major component is equally hydrophilic as hemicellulose. Also, there are different studies showing that lignin is amphiphilic, i.e., it has both hydrophophic and hydrophilic groups. Several studies have shown that lignin enhances the bonding with hydrophophic polymers.
The authors need to mention the semi crystallinity of cellulose, the fibrillar structure that enhances the reinforcement factor in polymer composites.
2) Need to cite some very relevant works such as “ Nanoscale characterization of natural fibers and their composites using contact-resonance force microscopy’ in introduction. This was the only study in which the mechanical property (stress) of interphase is natural fiber reinforced polymer composites were quantitatively characterized.
3) Page 3 Line 96-97. “ The fibers were conducted by sun-drying.”
Conducted? Make it clear
4) How did you determine the E(modulus) of fibers treated and untreated in the method section? Please add it your methods.
5) Page 5 Line 158-166 SEM morphology
Where is figure (5e) Alkali-treated for 8h Typha spp. Fibers? It is not there in the text.
How do you know that hemicellulose, lignin and wax were removed from surface? Do you have any references? Also, there are different theories on the location of hemicellulose and lignin in natural fibers. There are mostly in between the cellulose microfibrils in the cell wall. Make it clear?
Also, Figure 5 a and b looks much rougher than c and D. Need to show high resolution SEM pictures to make it clear. Also, please add the diameter change to the fibers after alkaline treatment. That gives more clarity.
6) For Figure 6 and Figure 8, need to show R square values for each scatter plots to show the amount of scatter for (a) matrix length, L, (b) fiber 173 spacing, t, (c) fiber diameter, D, and bonding area, A.
7) You need to explain why the matrix length and fibers increase interface toughness while the diameter and bonding area had negative effect.
8) Page 9 Line 234-236. “Figure 10 shows that the Typha spp. fiber/epoxy model composite has higher interfacial fracture values than Typha spp. fiber/PLLA model composite, this occurs because PLLA-based composite has lower mechanical properties compared to epoxy”.
Do you have the individual bulk mechanical property of PLLA and epoxy to arrive at this conclusion? There are different types of PLA which has better mechanical properties than epoxy and vice versa.
9)In Figure 15, fibers after 4 hours and 8h of treatment in epoxy composites showed lower interfacial shear strength than PLLA composites? We did not see this trend in Figure 10. Explain
Round 2
Reviewer 2 Report
Manuscript has significantly improved